# Deformation Behavior of a Double Soaked Medium Manganese Steel with Varied Martensite Strength

**Alexandra Glover [1],\*, Paul J. Gibbs [2], Cheng Liu [2], Donald W. Brown [2], Bjørn Clausen [2], John G. Speer [1] and Emmanuel De Moor [1],\***

[1]  Advanced Steel Processing and Products Research Center, Colorado School of Mines, Golden, CO 80401, USA

[2]  Los Alamos National Laboratory, Los Alamos, NM 87545, USA

\*  Correspondence: aglover@mymail.mines.edu (A.G.); edemoor@mines.edu (E.D.M.)

**Abstract:** The effects of athermal martensite on yielding behavior and strain partitioning during deformation is explored using in situ neutron diffraction for a 0.14C–7.14Mn medium manganese steel. Utilizing a novel heat treatment, termed double soaking, samples with similar microstructural composition and varied athermal martensite strength and microstructural characteristics, which composed the bulk of the matrix phase, were characterized. It was found that the addition of either as-quenched or tempered athermal martensite led to an improvement in mechanical properties as compared to a ferrite plus austenite medium manganese steel, although the yielding and work hardening behavior were highly dependent upon the martensite characteristics. Specifically, athermal martensite was found to promote continuous yielding and improve the work hardening rate during deformation. The results of this study are particularly relevant when considering the effect of post-processing thermal heat treatments, such as tempering or elevated temperature service environments, on the mechanical properties of medium manganese steels containing athermal martensite.

**Keywords:** neutron diffraction; austenite stability; medium manganese steel; double soaking; localized deformation

## 1. Introduction

Modern automotive designs, which desire increased passenger safety in conjunction with vehicle light-weighting and increased fuel efficiency, require the development of new advanced high strength steel (AHSS) grades [1–3]. The first generation of advanced high strength steels, which includes dual-phase, transformation-induced plasticity, and complex phase steels, are primarily ferrite-based. The second generation of AHSS utilize alloys with high manganese contents to generate fully austenitic microstructures such as twinning-induced plasticity (TWIP) steels. The high alloy contents needed to achieve complete austenite stabilization present processing challenges which, combined with the increased cost of these alloys, have delayed the application of TWIP steels in automotive applications. With these processing challenges in mind, the third-generation of AHSS have been proposed with the goal of generating higher strength–ductility combinations than the first generation of AHSS while using lower alloying element concentrations than the second generation of AHSS.

One of the new steel grades proposed to reach the property targets desired for the third generation of AHSS are medium manganese steels. These steels generally contain between 5–10 wt % Mn and 0.1–0.3 wt % C, and are intercritically annealed to form a microstructure of ferrite plus austenite [4–6]. With typical retained austenite volume fractions between 5 and 30 vol %, the improved mechanical properties of these steels can be attributed in part to the strain-induced transformation of austenite

to martensite, which has been shown to help maintain a high rate of work hardening during deformation [4,7–11]. In intercritically annealed medium manganese steels, the resistance of austenite to strain-induced transformation has been shown to be highly dependent upon intercritical annealing heat treatment parameters [5,6,12]. This is because the intercritical annealing temperature within the ferrite plus austenite phase field determines the equilibrium austenite volume fraction and the associated C and Mn enrichment.

These intercritically annealed medium manganese steels, which contain primarily retained austenite and ferrite, may be strengthened through the substitution of ferrite with athermal martensite. Several studies of medium manganese steels which contain small volume fractions of athermal martensite have shown, in some conditions, to promote continuous yielding and improve the overall strength–ductility combination of the material [5,6,12–14]. Due to the nature of the single intercritical annealing heat treatment used in these studies, the introduction of martensite was only possible through the incomplete stabilization of austenite during intercritical annealing. This approach leads to retained austenite with reduced levels of carbon and manganese partitioning, which may not be desirable.

The newly proposed double soaking heat treatment provides a unique opportunity to study the influence of martensite in medium manganese steels [15,16]. This is because the double soaking heat treatment allows for a large volume fraction of martensite with variable strength to be generated while maintaining high volume fractions of retained austenite with significant manganese partitioning. Previous research has shown that the double soaking heat treatment is capable of generating ultimate tensile strengths above 1600 MPa in conjunction with total elongations above 13% in a 0.14C–7.14Mn steel [16]. In this study, the response of a 0.14C–7.14Mn steel to the double soaking or double soaking plus tempering heat treatment was examined using in situ neutron diffraction. This allowed for the role of martensite strength and characteristics, such as dislocation density, in yielding behavior and during plastic deformation to be studied.

## 2. Materials and Methods

The double soaking heat treatment is shown schematically in Figure 1. The first step, the primary soaking treatment in the intercritical region, is characterized by primary austenite ($\gamma_P$) nucleation and growth and manganese partitioning from ferrite to austenite. Next, a short secondary soaking treatment is applied at a higher temperature as either a continuous or discontinuous heat treatment. The microstructure at the secondary soaking temperature consists of primary austenite present after the initial soak ($\gamma_P$), newly formed secondary austenite ($\gamma_s$), and (potentially) some remaining ferrite ($\alpha$), depending upon the selected secondary soaking temperature and time. During the secondary soaking treatment, a substantial amount of manganese diffusion from the enriched $\gamma_P$ to the newly formed $\gamma_s$ is not desired and may be avoided given the slow rate of manganese diffusion in austenite combined with the short secondary soaking time. Depending upon the time and temperature of the secondary soaking treatment, varying amounts of carbon diffusion are expected. Finally, the steel is quenched to room temperature, which is below the martensite start ($M_s$) temperature of $\gamma_s$, causing the austenite formed during the secondary soaking operation to transform into martensite. This results in a final microstructure of retained austenite, martensite, and (potentially) intercritical ferrite. An optional tempering treatment, in this instance at temperatures above the $M_s$, may also be added to modify martensite strength. This additional heat treatment is classified as the double soaking plus tempering (DS-T) heat treatment in this investigation.

This study explored the influence of martensite strength on austenite stability using a steel with the composition 0.14C–7.17Mn–0.21Si (wt %). This steel was cold-rolled to a final thickness of 1.4 mm and then intercritically annealed by industrial batch annealing, which constitutes the primary soaking treatment. Following the primary soaking treatment, the $A_{c1}$ and $A_{c3}$ temperatures were measured at 668 and 825 °C, respectively, and the microstructure was found to consist of 40 vol % austenite and 60 vol % ferrite using X-ray diffraction. The tensile properties generated following the primary soaking heat treatment are shown in Figure 2. The primary soaked material was characterized as

having an ultimate tensile strength of 1038 MPa combined with a total elongation of 41%. The material underwent localized deformation upon yielding, with the yield point drop and yield point elongation characteristic of many medium manganese steels. This was followed by a region of moderate work hardening and limited post-uniform elongation.

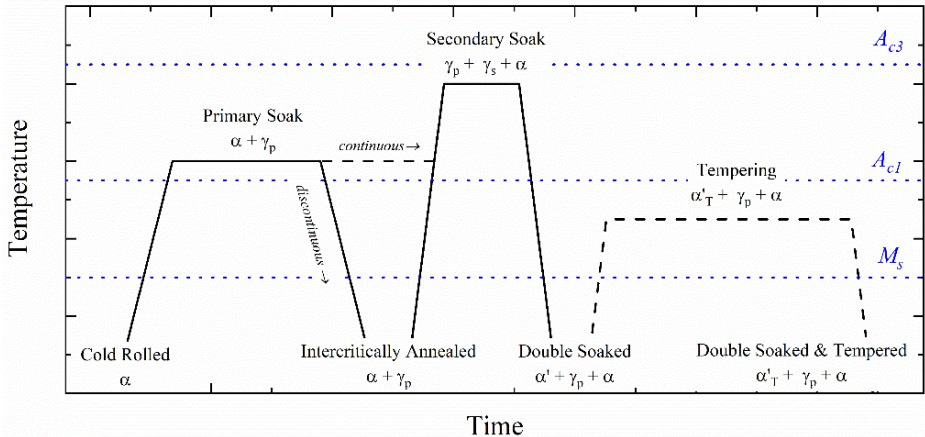

**Figure 1.** Schematic of the double soaking and double soaking plus tempering heat treatment. Expected changes in microstructure are indicated: Ferrite ($\alpha$), primary austenite ($\gamma_P$), secondary austenite ($\gamma_s$), martensite ($\alpha'$), and tempered martensite ($\alpha'_T$).

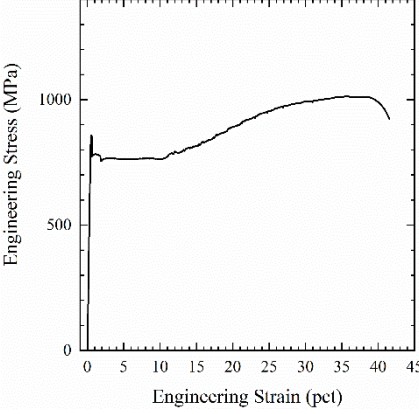

**Figure 2.** Room temperature, quasi-static engineering stress-strain curve for the industrially batch annealed (primary soaked) material.

In this study, two experimental heat treatments were then applied to the batch annealed (BA) material. These heat treatments were designed to generate equivalent volume fractions of martensite while varying martensite strength. The first heat treatment, called the double soaking (DS) treatment, involved the application of the secondary soaking heat treatment at 800 °C for 30 s to the BA steel [16]. The second heat treatment, identified as the double soaking plus tempering (DS-T) treatment, added both a secondary soaking heat treatment at 800 °C for 30 s followed by a quench to room temperature as well as subsequent application of a tempering treatment at 450 °C for 300 s. Both heat treatments were applied using salt pots, with an average heating rate of 80 °C/s.

Uniaxial tensile testing was conducted using ASTM E8 subsized samples machined parallel to the rolling direction with a gauge section of 25 mm. All tests were performed on a Instron 1125 (Noorwood, MA, USA) screw-driven loading frame at a constant engineering strain rate of $4.5 \times 10^{-4}$ s$^{-1}$ in a method consistent with ASTM E8 [17]. Applied load and crosshead displacement were recorded at a rate of 10 Hz. Sample deformation was monitored using 2D digital image correlation (DIC), a technique which relies on the computer vision approach to extract whole-field displacement data [18]. Images

were captured using a CCD camera with a resolution of 2448 × 2048 pixels and a lens with a 75 mm focal length. The physical resolution was 22.5 μm. Images were taken at a frequency of 5 Hz. Before testing, a random pattern was printed onto each specimen surface by first spraying white paint as a background and then black paint to generate random speckles. Using the displacement vectors reported by the VIC 3D software developed by Correlated Solutions (Irmo, SC, USA) and a virtual 25.4 mm (1 in) extensometer, the macroscopic strain for each sample was characterized [19].

In situ neutron diffraction was performed on the SMARTS instrument at the Lujan Center at Los Alamos National Laboratory [20]. Phase fractions, elastic lattice strains, and diffraction peak width measurements were made during tensile deformation using two banks of detectors oriented at ± 90° to the incident beam. One detector bank collected data from the transverse direction (normal to the specimen thickness), and the other collected data from the axial direction (along the axis of maximum tension). Tensile specimens, with a 44.5 mm reduced gauge section and a 6.4 mm gauge width, were deformed incrementally at a constant engineering strain rate of $4.5 \times 10^{-4}$ s$^{-1}$, and diffraction patterns were recorded while the sample was held at a constant displacement for approximately 1800 s [17]. The reported stress values were taken as the final value measured during each diffraction hold, and the magnitude of stress relaxation measured during each diffraction hold varied between 20 and 60 MPa following macroscopic yielding for both heat treatment conditions. Sample deformation was monitored using 2D DIC during the in situ neutron diffraction. Due to the increased length of the interrupted neutron diffraction tests, the sampling interval was reduced to 0.0034 Hz. Using the displacement vectors reported by the VIC 3D software developed by Correlated Solutions and a virtual 25.4 mm (1 in) extensometer, the macroscopic strain for each sample was determined [19].

Two phases were considered in the analysis of the diffraction data—body centered cubic (BCC) ferrite/martensite (α/α′), and face centered cubic (FCC) austenite (γ). The alpha prime (α′) martensite fraction in the starting microstructure was estimated from dilatometric scans, as martensite tetragonality was insufficient to differentiate body centered tetragonal (BCT) martensite from BCC ferrite in the diffraction data [21]. Austenite fractions were determined at each deformation step using the Rietveld analysis function within the GSAS software package [22].

Single peak fitting using the Rawplot subroutine of GSAS was used to measure changes in interplanar spacing at each deformation step where diffraction data were collected. Elastic lattice strain ($\varepsilon_{hkl}$), a measurement of elastic strain partitioning between phases, was calculated using Equation (1):

$$\varepsilon_{hkl} = \left( d_{hkl}^{\sigma} - d_{hkl}^{0} \right) / d_{hkl}^{0} \tag{1}$$

where $d_{hkl}^{\sigma}$ is the interplanar spacing for a set of lattice plane normals, parallel to the diffraction vector measured under an applied stress, and $d_{hkl}^{0}$ is the interplanar spacing in the stress-free condition. Stress-free interplanar spacings were measured by extrapolating the linear-elastic region of the true stress-lattice strain curve to a zero-load condition, which should have removed any effects of sample/fixture unbending. The reported elastic lattice strain values were all taken from the axial detector bank.

## 3. Results

The initial phase fractions for the specimens with the DS and DS-T heat treatments are shown in Table 1 along with the heat treatment parameters applied to each sample. The austenite phase fractions were measured during the initial diffraction measurement in the axial direction at a stress of 40 MPa based upon full Rietveld full pattern refinements. The athermal martensite phase fractions were estimated through the application of the lever rule to dilatometric data in a method consistent with Kang et al. [23]. Ferrite was assumed to compose the remaining balance of the microstructure, as no other phases were identified in the diffraction patterns of the unstrained samples.

**Table 1.** Heat treatment parameters and resulting phases fractions.

| | **Heat Treatment** | $\gamma$ | $\alpha'$ | $\alpha$ |
|---|---|---|---|---|
| **Sample ID** | **Heat Treatment Parameters** | **vol %** | **vol %** | **vol %** |
| DS | 800 °C, 30 s | 28 | $60\left(\alpha'_F\right)$ | 12 |
| DS-T | 800 °C, 30 s + 450 °C, 300 s | 29 | $60\left(\alpha'_T\right)$ | 11 |

Engineering stress-strain curves for the DS and DS-T conditions are shown in Figure 3a,c, respectively. The DS condition exhibited continuous yielding, followed by a region with a high rate of work hardening, and no post-uniform elongation. This resulted in an ultimate tensile strength of 1710 MPa and total elongation of 12.8% for the DS sample tested in uniaxial tension. The corresponding quasi-static and interrupted neutron diffraction true stress-true strain tensile curves are shown in Figure 3b,d. The neutron diffraction curve, which shows the true stress-true strain location of each incremental step where diffraction data were collected, had reduced true stress values as compared to the corresponding true strain value on the quasi-static curve. Additionally, the total elongation of the neutron diffraction sample was reduced as compared to the sample tested under quasi-static conditions in uniaxial tension. Overall, the tensile curve characteristics of interest, including the high rate of work hardening and continuous yielding behavior, were maintained in the sample that was incrementally deformed during neutron diffraction.

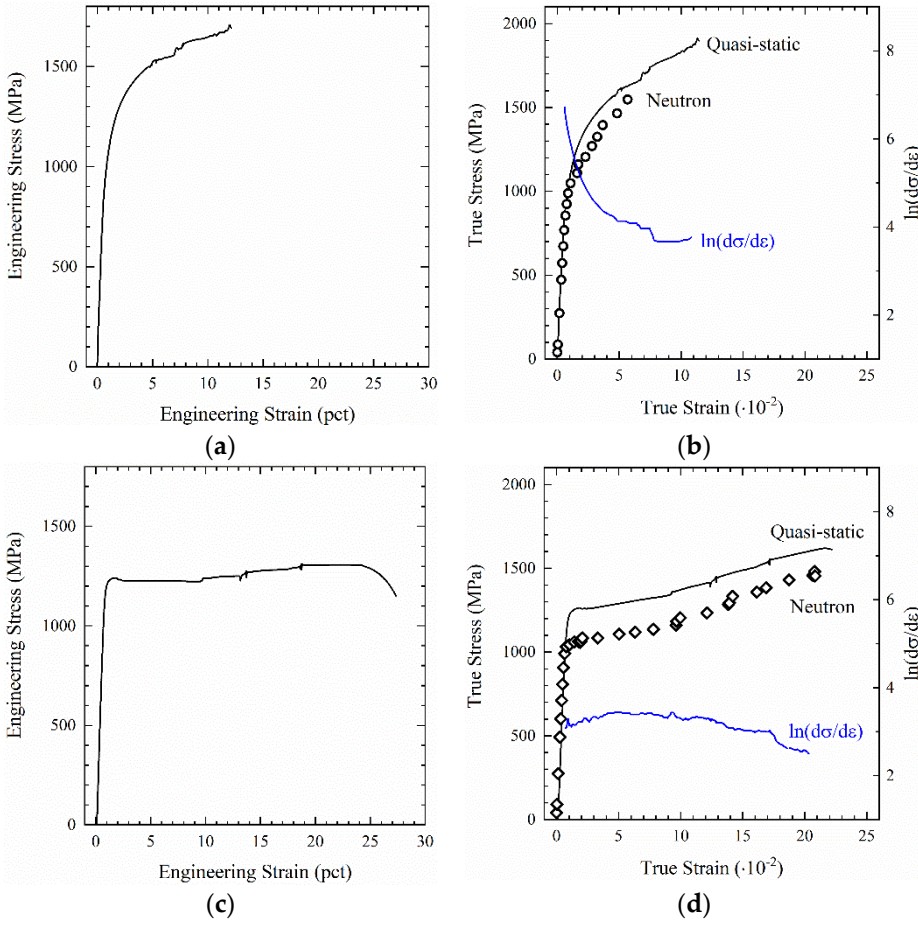

**Figure 3.** Room temperature engineering and true stress-strain curves for the (**a,b**) double soaked (DS) and (**c,d**) double soaked and tempered (DS-T) samples tested in both quasi-static and in situ neutron diffraction testing conditions along with the instantaneous work hardening rate, $\ln(d\sigma/d\varepsilon)$.

Representative indexed diffraction patterns, measured at a range of true stress values in the axial orientation for the DS and DS-T heat treatment conditions, are shown in Figure 4a,b respectively. The phase fractions of only two phases—BCC ferrite/martensite and FCC austenite—could be quantified. Small diffraction peaks from the {101} orientation of hexagonal close packed (HCP) epsilon martensite ($\varepsilon$) were evident in the diffraction patterns for both DS and DS-T samples at increased stress levels, although the intensity was not sufficient to allow the epsilon martensite volume fractions to be estimated.

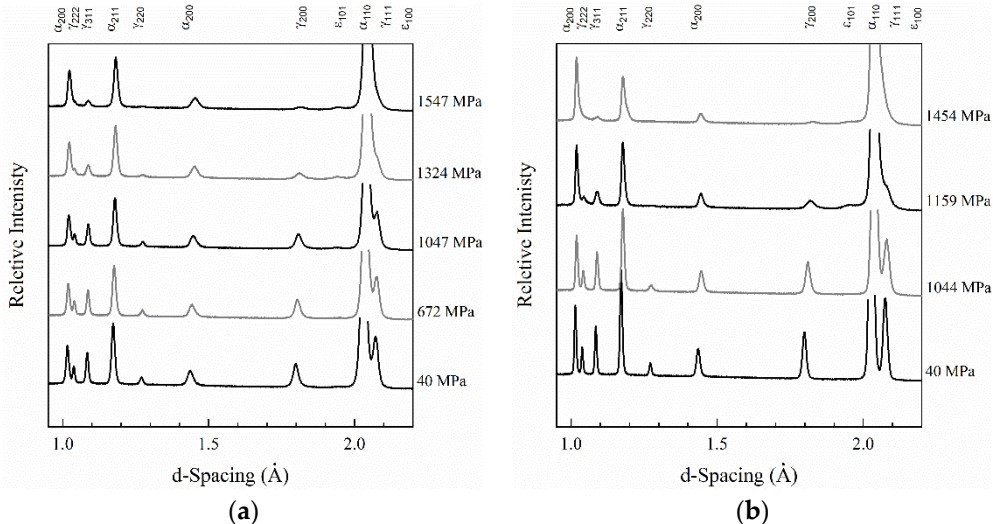

**Figure 4.** Representative in situ neutron diffraction data during uniaxial deformation for the (**a**) double soaked and (**b**) double soaked plus tempered heat treatment conditions. True stresses at each diffraction condition are indicated.

Using the diffraction patterns collected during each deformation step, the austenite fractions were calculated for both the DS and DS-T samples. The calculated austenite fractions are plotted as a function of true stress in Figure 5, and, in all cases, the calculated uncertainty was less than ±0.12 vol %. In the unstrained condition, the DS sample, shown in Figure 5a, contained 28 vol % austenite. Upon macroscopic yielding, the austenite transformation occurred at a nearly constant rate with increasing stress. The final austenite volume fraction measured before fracture was 7.5 vol %. In the DS-T condition, the strain-induced transformation of austenite occurred primarily during the yield point elongation (YPE), during which approximately 12 vol % of austenite transformed. In the region of limited work hardening that follows YPE, the strain-induced transformation of austenite slows significantly with only 4 vol % austenite transformed before sample failure. In both the DS and DS-T heat treatment conditions retained austenite volume fractions decreased by less than 1 vol % before macroscopic yielding was observed. This may indicate that a small fraction of stress-assisted austenite transformation occurred. Due to the small volume fraction (potentially) transformed by the stress-assisted mechanism, the resulting impact on tensile properties was likely insignificant.

Figure 6 shows the evolution of elastic lattice strains for the {211} orientation of the BCC phases ($\alpha/\alpha'$) and the {311} orientation of FCC austenite as a function of macroscopic true strain measured in the axial orientation. By applying an isostrain assumption ($\sigma = E \cdot \varepsilon$) the elastic lattice strain values shown in Figure 6 can be assumed to be proportional to an equivalent stress [24]. The elastic lattice strain values calculated for the DS condition are shown in Figure 6a, and the values for the DS-T condition are shown in Figure 6c along with the austenite volume fractions calculated at each deformation step. In Figure 6b,d, the elastic lattice strains and austenite volume fractions measured at low macroscopic true strain values are magnified for both the DS and DS-T sample conditions. In all cases, error bars were omitted, as they were smaller than the data points. Additionally, the calculated linear fit for the linear elastic region of the $\gamma$ and $\alpha$ elastic lattice strains are plotted, as they are helpful in identifying the order in which individual phases yield. In all plots of elastic lattice strain, the {211}

orientation of the BCC phases ($\alpha/\alpha'$) and the {311} orientation of austenite are displayed, as they have been shown to have the most linear lattice strain response, thus making them the best orientations to use when characterizing the macroscopic stresses or strains for a sample tested in uniaxial tension with limited texture [25].

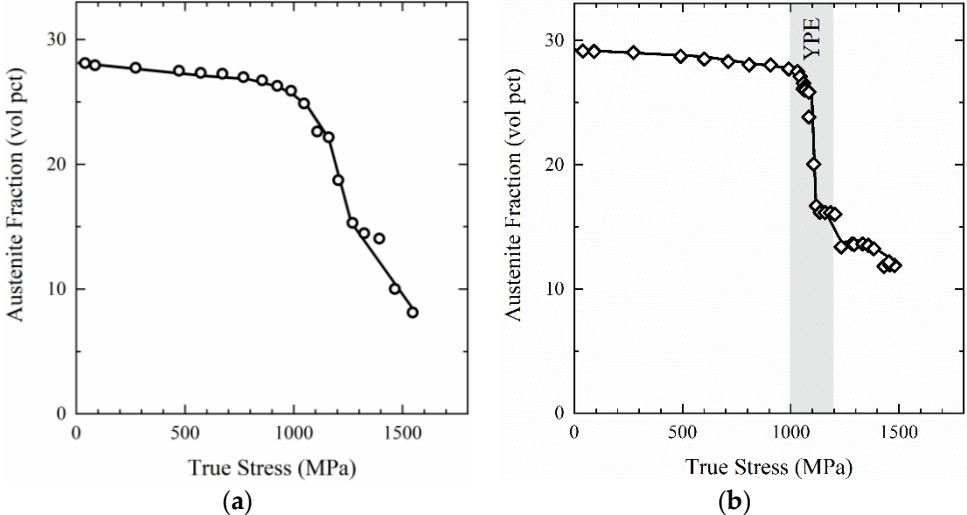

**Figure 5.** Austenite volume fractions as a function of macroscopic true stress for the (**a**) double soaked (DS) and (**b**) double soaked and tempered (DS-T) heat treatment conditions.

In the DS condition, three distinct deformation stages can be identified from the elastic lattice strain data, which is plotted as a function of macroscopic true strain in Figure 6b. In stage I, linear elastic (reversible) deformation occurred in both $\alpha/\alpha'$ and $\gamma$. Stage II was initiated at a macroscopic true strain value of 0.0044 and a corresponding estimated macroscopic true stress of 626 MPa when $\gamma_{\{311\}}$ deviated from linear elastic loading, as shown in Figure 6b. This deviation from reversible elastic loading indicates the initiation of plastic deformation in $\gamma$. Based upon the corresponding austenite volume fractions plotted in Figure 6b, the transformation of significant volume fractions of austenite has not yet occurred. Finally, stage III was initiated when yielding occurred in the $\alpha/\alpha'$ matrix at a macroscopic true strain of 0.0056 and corresponding macroscopic true stress of 768 MPa. At the onset of stage III, significant volume fractions of austenite were retained, although the strain-induced transformation of austenite began as indicated by the decrease in austenite fractions with increasing macroscopic true strain, clearly seen in Figure 6a. With an increasing macroscopic true strain, the elastic lattice strain (which is directly proportional to stress) in the BCC phases continued to increase until the neutron diffraction sample fractured at a macroscopic true strain of 0.056.

In the elastic lattice strain data for the DS-T sample, shown in Figure 6c, three stages of deformation can also be identified. Following the reversible elastic deformation of $\alpha/\alpha'$ and $\gamma$ in stage I, stage II was initiated by yielding in austenite at a macroscopic true strain of 0.0034 and corresponding estimated macroscopic true stress of 624 MPa, as indicated in Figure 6d. Following the initiation of stage II, stress in both $\gamma$ and $\alpha/\alpha'$ continued to increase as shown by the increasing elastic lattice strains in both phases. During stage II, austenite fractions remained nearly constant, indicating that the strain-induced transformation of austenite was not yet occurring. The yielding of the $\alpha/\alpha'$ phase, at a macroscopic true strain of 0.0041 and corresponding macroscopic true stress of 718 MPa, initiated stage III. During stage III, stress levels in austenite were relatively constant and austenite fractions began to decrease. The most rapid rate of austenite transformation occurred during the YPE, as identified in the macroscopic true-stress, true-strain curve, after which the transformation rate slowed significantly. The corresponding elastic lattice strain for the $\alpha/\alpha'$ phase increased slightly throughout stage III with increasing macroscopic true strain, indicating increasing stress in the $\alpha/\alpha'$ phase.

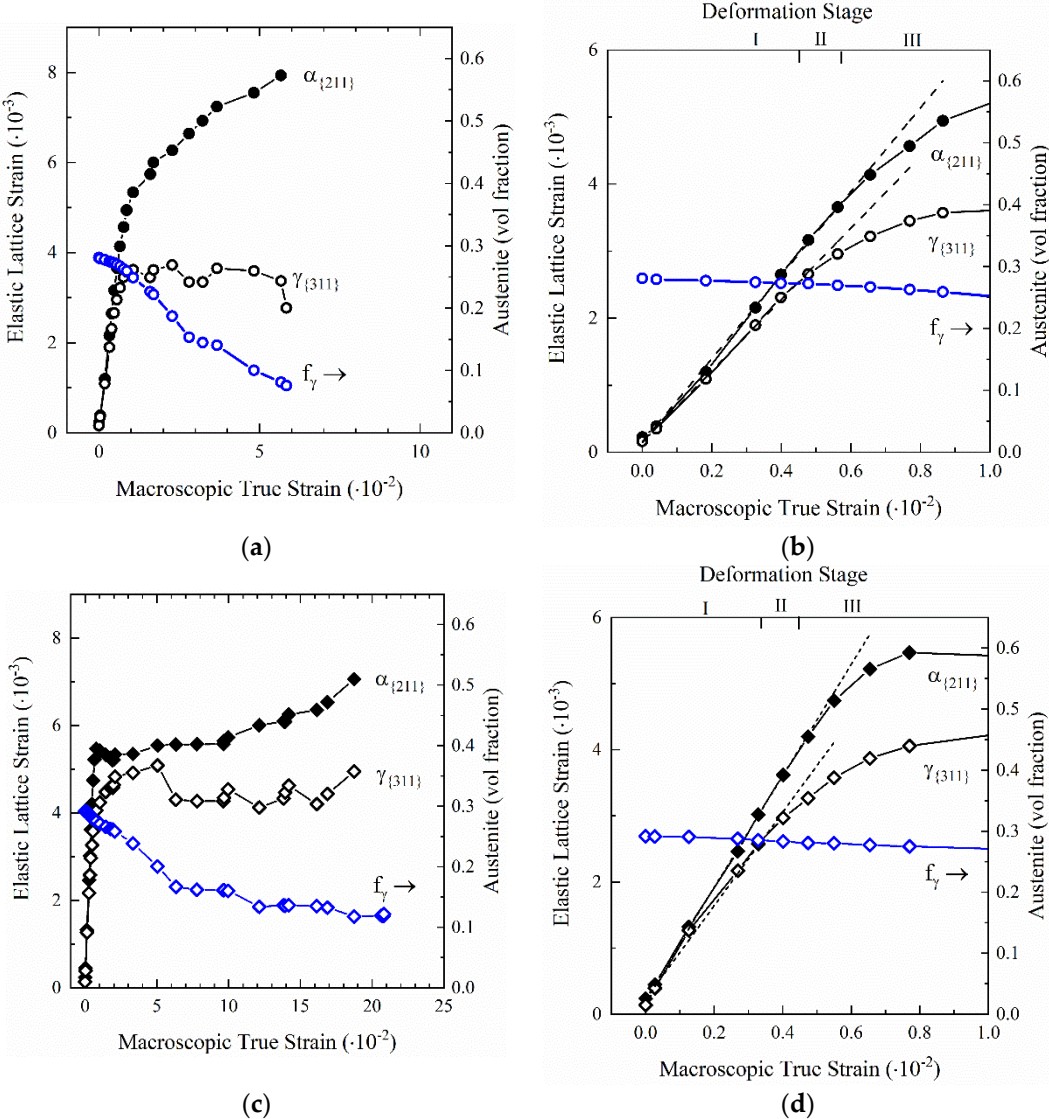

**Figure 6.** Elastic lattice strain as a function of macroscopic true strain for the {211} orientation of the BCC phases ($\alpha/\alpha'$) and {311} orientation of austenite for the (**a**,**b**) DS and (**c**,**d**) DS-T heat treatment conditions.

## 4. Discussion

The heat treatments presented in this work were selected to generate microstructures containing the same amount of retained austenite with similar strength while varying the strength and characteristics of the matrix phase. These modifications were achieved through the application of the double soaking treatment with an added tempering step. This modified the strength and mobile dislocation density along with other microstructural characteristics of the athermal martensite, which composed the bulk for the matrix phase. These differences were seen to significantly influence true stress-true strain curves (Figure 3)—the yielding behavior and evolution of the incremental work hardening rate during deformation, in particular.

### 4.1. Role of Martensite in Yielding Behavior

The results of this study suggest that the yielding behavior of medium manganese steels is highly dependent upon the characteristics of the matrix phase. Significant changes in yielding behavior were observed between the sample containing as-quenched athermal martensite (DS heat treatment) and the sample containing tempered athermal martensite (DS-T heat treatment). Specifically, during

the deformation of the initial double soaked microstructure, which contained large fractions of as-quenched athermal martensite, continuous yielding was observed. With the addition of a tempering treatment, which was likely accompanied by a reduction in dislocation density and precipitation of carbides [26–29], the discontinuous yielding traditionally seen in medium manganese steels was observed in the macroscopic true-stress, true-strain curve. Additionally, the load drop and YPE-like behavior observed in the $\alpha_{\{211\}}$ elastic lattice strain data shown in Figure 6c would be consistent with the pinning of mobile dislocations by carbon segregation and carbide precipitation in martensite during tempering, as proposed by Krauss and Swarr [27].

The results of this study support the findings of Steineder et al. and De Cooman et al., both of whom suggested that the introduction of a large volume fraction of mobile dislocations due to the formation of athermal martensite was effective at preventing discontinuous yielding in medium manganese steels [14,30]. The prevention of discontinuous yielding through an increase in mobile dislocation density has also been shown effective in studies which utilized pre-straining to increase mobile dislocation density [31]. Other proposed mechanisms for the elimination of discontinuous yielding in medium manganese steels, such as the modification of austenite strength and volume fractions or the sequential deformation of individual phases during yielding, were not found to influence the yielding behavior of the medium manganese steel used in this study, as these factors were held constant between the two sample conditions examined [13,32,33].

### 4.2. Role of Martensite during Plastic Deformation

The role of martensite during plastic deformation is evident through a comparison of the macroscopic tensile properties shown in Figure 3 and the elastic lattice strains shown in Figure 6. The DS condition, with a matrix of primarily as-quenched athermal martensite, maintains a high rate of work hardening following macroscopic yielding. Through an examination of the elastic lattice strain data for the corresponding heat treatment, it is evident that the rapidly increasing elastic lattice strains in the $\alpha/\alpha'$ phase in stage III due to work hardening and the dynamic replacement of austenite with martensite is responsible for the high work hardening rate seen in the macroscopic true stress-true strain curve. The corresponding elastic lattice strains for the $\gamma$ phase remain nearly constant during stage III, as the stress in austenite is limited by the transformation criteria. Overall, the elastic lattice strain data for the DS condition indicate that the high rate of work hardening can be attributed to work hardening in the $\alpha/\alpha'$ matrix and the strain-induced transformation of austenite to martensite.

In the DS-T condition, where the matrix was primarily composed of tempered athermal martensite, the macroscopic true stress-true strain curve shows that the ultimate tensile strength and work hardening rate decreased as compared to the DS condition. The elastic lattice strain data and austenite fractions shown as a function of true strain in Figure 6 are helpful in understanding the contribution of martensite to deformation behavior following yielding. The similar increase in stress partitioned to the $\alpha/\alpha'$ phase during stage III can be attributed to two factors. The first factor to consider is the limited work hardening response of tempered martensite, which composes the majority phase in the matrix. Studies of low-carbon martensite tempered at intermediate temperatures (300–450 °C) for similar times have shown that martensite tempered in this regime generally produces an extremely limited work hardening response during deformation [27–29,34,35]. This low rate of work hardening has been attributed to the replacement of the cellular dislocation structure found in as-quenched martensite, with a non-cellular distribution of dislocations which by-pass carbides formed during tempering, leading to a limited dislocation interaction and accumulation [27,34]. Secondly, the reduced rate of austenite transformation following the macroscopic YPE limits the fraction of martensite added to the matrix. This reduces the overall strength of the material at high macroscopic strains, as lower strength austenite has not been transformed to higher strength martensite. Overall, the reduced work hardening response of the tempered athermal martensite composing the majority of the matrix and the reduced rate of austenite transformation following YPE lead to the overall reduced work hardening rate seen in the macroscopic true stress-true strain curve.

## 5. Conclusions

The role of martensite characteristics and strength in a 0.14C–7.14Mn steel was explored using a novel double soaking heat treatment. The double soaking process utilizes a two-step heat treatment where the primary soak generates a significant volume fraction of retained austenite stabilized by carbon and manganese partitioning, and the secondary soak substitutes athermal martensite for ferrite in the matrix. In this study, the double soaking heat treatment was applied to generate microstructures with equivalent volume fractions of austenite and martensite while varying martensite strength. Using in situ neutron diffraction to study the role of athermal martensite in yielding and plastic deformation, it was shown that:

1.  The inclusion of large volume fractions of as-quenched athermal martensite in the microstructure of medium manganese steels promotes continuous yielding in both the FCC and BCC/BCT phases and in the overall macroscopic tensile curve. Following tempering, discontinuous yielding was found to occur in both the BCC/BCT phase and in the overall macroscopic tensile curve. The shift from continuous to discontinuous yielding is believed to be linked to a reduction in mobile dislocation density in the BCC/BCT phase as a result of the tempering heat treatment.
2.  The work hardening rate of a medium-manganese steel is highly dependent upon the properties of the matrix. An initial matrix composed primarily of as-quenched athermal martensite is effective in producing a sustained high rate of work hardening, primarily due to work hardening in the martensitic phase and the dynamic strain-induced transformation of austenite to martensite.
3.  The substitution of ferrite with martensite (as-quenched or tempered) in the initial microstructure of a medium manganese steel improves the strength–ductility product of the material.

**Author Contributions:** Conceptualization, formal analysis, and writing—original draft preparation A.G.; conceptualization, methodology, and writing—review and editing, E.D.M., J.G.S., and P.J.G.; methodology, formal analysis, resources, and data curation, C.L., D.W.B., B.C.

**Funding:** The sponsors of the Advanced Steel Processing and Products Research Center (ASPPRC) at the Colorado School of Mines are gratefully acknowledged. This work benefited from the use of the Lujan Neutron Scattering Center at LANSCE, LANL. Los Alamos National Laboratory is operated by Triad National Security, LLC, for the National Nuclear Security Administration of U.S. Department of Energy (Contract No. 89233218CNA000001).

**Conflicts of Interest:** The authors declare no conflict of interest. The funders had no role in the design of the study; in the collection, analyses, or interpretation of data; in the writing of the manuscript, or in the decision to publish the results.

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
