# Peer review of "Deformation Behavior of a Double Soaked Medium Manganese Steel with Varied Martensite Strength"

_metals, doi:10.3390/met9070761_

Round 1

Reviewer 1 Report

This is an interesting study which merits publication after the following questions/comments are addressed (with apologies for the random ordering):

1.       Please provide the essential processing details for the experimental steels including all annealing temperatures, annealing times, measured or calculated Ac1/Ac3 values, cold rolling reduction and final sheet thickness.

2.       Was the experimental steel allowed to cool to RT after the primary soak? If so what were the retained austenite and martensite fractions at this point?

3.       It would be helpful to show the stress-strain curves for the material just after the primary soak.

4.       DIC tensile data is alluded to but not exploited in this article – this is a pity as strain localisations are present.

5.       The tensile curve of Figure 2a looks brittle – does the material fracture before the Considere criterion?

6.       The conditions of the neutron diffraction experiment are not clearly described. How was the macroscopic “neutron” strain measured? Were the lattice elastic strains in the directions parallel and perpendicular to the tensile axis recorded simultaneously? If so then the principal stresses could be calculated from the lattice strains (assuming in-plane biaxial stress conditions) and a comparison made with the macroscopic value. This would give a better insight into the stress partitioning between the fcc and bcc phases. Please comment.

7.       What was the magnitude of any stress relaxation effects observed during the neutron diffraction experiment?

8.       Please convert all austenite contents from weight fraction to volume or phase fraction.

9.       In Table 1 please provide a comparison between the austenite fraction measured by dilatometry and that measured (at zero strain) by neutron diffraction.

10.   I strongly agree that {211}a and {311}g are the only realistic reflections to use.

11.   Line 87 focus=focal

12.   Line 232 {311} = {211}

13.   Assuming E311g =183 GPa then the transformation stress of the DS RA is ~ 660 MPa and that of DS-T RA is ~770 MPa i.e. tempering has apparently increased the austenite stability. Can the authors comment?

Reviewer 2 Report

The article presents an topic with an investigation about considering the effect of post-processing thermal heat treatments such as tempering or elevated temperature service environments on the mechanical properties of  medium manganese steels containing athermal martensite., however there are points that need to be clarified and summarized.

1. Introduction: The introduction is very general. It should be improved and clearly explain the purpose and final objective of of this study with appropriate references. From my point of view, introduction is not well focused. In a research paper, it is expected that introduction section briefly explains the starting background and, even more important, the originality (novelty) and relevancy of the study is well established. Once this is done, hypothesis and objectives of the study need to be addressed, as well as a brief justification of the conducted methodology.I think that figure 1 ("Schematic of the double soaking and double soaking plus tempering heat treatment") should go to the section 2 "materials and methods"

2. Results and Discussion Section: There are no micrographs in this section.  From my point of view, micrographs that reflect in a visual way the previous and subsequent results of the applied actions must be introduced in the article, commenting clearly on how the structure of the material varies.

3. Section Conclusions: Improve the conclusions section (is very short), taking into account what is described in the results section.

Round 2

Reviewer 2 Report

I agree with the changes made by the authors